# Hygroscopic Properties of Three Cassava (*Manihot esculenta* Crantz) Starch Products: Application of BET and GAB Models

**DOI:** 10.3390/foods11131966

**Published:** 2022-07-02

**Authors:** Aneta Ocieczek, Dominika Mesinger, Henryk Toczek

**Affiliations:** 1Faculty of Management and Quality Science, Gdynia Maritime University, 81-225 Gdynia, Poland; d.mesinger@sd.umg.edu.pl; 2Faculty of Marine Engineering, Gdynia Maritime University, 81-225 Gdynia, Poland; h.toczek@wm.umg.edu.pl

**Keywords:** tapioca, hygroscopicity, *BET* and *GAB* models, particle size characteristics, storage stability

## Abstract

This study aimed to compare hygroscopicity properties of three cassava (*Manihot esculenta* Crantz) products: native starch powder (NS), fermented starch powder (FS), and starch granulate (SG). The analyzed properties were compared based on the statistical evaluation of differences in the course of sorption isotherms and the identification and comparison of parameters in two theoretical models of sorption. Empirical data were generated by means of the static-desiccator method. Measurements were made using AquaLab apparatus. The size, shape, and number of tapioca particles were characterized using a Morphology automatic particle analyzer. The study demonstrated that in-depth exploration of empirical data describing hygroscopicity of samples with the use of mathematical tools allows evaluating their physical parameters. The results obtained were analyzed in terms of correlations between physical and physicochemical properties determining utility traits of cassava starch. The NS and SS featured significantly higher hygroscopicity than SG, as evidenced by the values of all parameters analyzed in this study. The study results provided new information related to the management of the production process, safety, and stability of these products.

## 1. Introduction

Cassava (*Manihot esculenta* Crantz), belonging to the *Euphorbiaceae* family, is a highly popular crop in the tropical zones of the globe, likewise, e.g., wheat in European countries [1]. It is one of the most often cultivated crops worldwide, providing starch for foodstuff and feedstuff production [2,3]. It is of particular importance to the economies of such countries as Nigeria, Thailand, Ghana, Brazil, and Indonesia, as it plays the role of a strategic raw material [4,5,6]. In the years 2000–2018, its production increased by approximately 100 million tons only in the countries claimed to be its main producers [7], which points to the growing demand for this crop.

Cassava (*Manihot esculenta* Crantz) roots are used to manufacture various starch products with a different fraction size composition, the common feature of which is a high dehydration degree. Water content reduction enables a radical decrease in water activity, and by this means, achieves enzymatic and microbiological stability, which, in turn, ensures adequate storage stability [8,9]. This form of preservation and storage of cassava products is driven by its higher susceptibility to spoilage compared to other tuber roots (potato, sweet potato, yam) [10]. Consequently, the cassava root, which is a storage organ, like other tuberous organs, does not fall into dormancy even under favorable conditions [11]. Granulated products obtained from cassava starch in the form of uneven, polyhedral, or spherical granules are called tapioca [12,13].

Tapioca is becoming more and more popular and is also used in European countries. There are three basic forms of this starch available on the market: starch powder, starch granulate, and fermented starch powder produced in the lactic fermentation process. Fermented starch is used as an additive to some baked products, both sweet and dry ones. In Ghana, fermented starch is used to produce six groups of products traditionally consumed in African countries, including: fermented pulps—fufu and akpu; fermented flours—lafun, kanyanga, and luku; smoked fermented balls—kumkum and pupuru; steamed and fermented chips—chikwangue and ntuka; baked grits—gari, agbelima, kapok, and popogari; steamed grits—attieke.

Lactic acid fermentation of edible cassava roots is extremely important as it removes cyanogenic glycosides, linamarin, and lotaustralin, which are present in the plant in various proportions. Cyanogenic glycosides are toxic because their hydrolytic degradation releases hydrogen cyanide. Therefore, the fermentation of cassava is strongly recommended [14].

The food industry extensively uses cassava starch (tapioca) in each of the three mentioned forms. However, starch powder and granules are the most common in the European market, whereas fermented starch cannot be commonly purchased in Poland yet. Tapioca starch (non-fermented) is used as, e.g., a stabilizer or a thickener due to its favorable technological properties and very neutral or utterly neutral taste. Tapioca-based food products are recommended to persons susceptible to allergens, especially those allergic to wheat and/or maize proteins. In addition, the starch of this type is increasingly often applied in food products intended for children, owing to its beneficial physical properties, texture, stability, and neutral taste. However, the most popular food products available on the European market are Asian-style noodles, Asian-style crispy bread, gluten-free bread, or the very popular molecular caviar (used in the production of Bubbletea, i.e., tapioca starch in the form of balls soaked in a fruit-flavored syrup and added in this form to lemonade or tea). Tapioca can also be used to prepare, e.g., pancakes, cakes, or puddings [15,16,17].

Edible cassava starch, i.e., tapioca, is also used in pharmaceutical products as an excipient and a filling substance. In the native form, it is a constituent of many pharmaceutical formulations—a binder, disintegrant, and a diluent of the active substance. In turn, the modified tapioca starch has an even more comprehensive range of applications, depending on the type of modification and intended use, etc. [18].

Carbohydrates are the most important and, at the same time, the dominant component of each starch, determining its technological usefulness as well as the stability of functional properties and consumer safety. In particular, attention is paid to the ratio of amylose to amylopectin, which in tapioca is at the level of 17–24% amylose and 76–83% amylopectin [19]. The ratio of these fractions may be necessary for shaping the sorption, including hygroscopicity. The crystalline form of granules of the dominant component of the cassava root causes the water present in it to be characterized by a high level of its activity, amounting to approximately 0.85 [20]. At the same time, it should be emphasized that the degree of its processing affects both the water content and its activity. At the same time, there are reports showing that the natural variety of cassava and the degree of its processing generally have a negligible or unclear effect on sorption behavior [21]. This statement had to be verified in the light of the growing interest in cassava products on the European market.

This study aimed to compare the sorption properties, in terms of hygroscopicity, of three cassava (*Manihot esculenta* Crantz) starch products (native, fermented, granulated), by comparing their sorption isotherms and the parameters of selected mathematical models used to describe these isotherms. The use of the *BET* (Brunauer, Emmett, and Teller) and *GAB* (Guggenheim, Anderson, and De Boer) models was conditioned by their theoretical nature, allowing for a physical interpretation of the parameters of these models. Moreover, its goal was to identify selected parameters describing the microstructure of the tested products. The study also assessed the differentiation of particle size and shape distributions of the studied starch samples. It was assumed that the analysis of the differentiation of the hygroscopic properties of starch, taking into account the variability of selected physical parameters of these starch particles, will be a source of new valuable information for the management of the production process, safety, and stability of these products.

## 2. Material and Methods

### 2.1. Materials

Three products made of cassava (*Manihot esculenta* Crantz) starch, also called tapioca, were studied: native starch powder (NS), starch granulate (SG), and fermented starch powder (FS). All three starches were produced under industrial conditions. Native tapioca powder (NS) and tapioca granulate (SG) were purchased in Poland, whereas fermented starch (FS) was imported from Brazil. Native tapioca powder was produced in Thailand and distributed in Poland by De Care Group Sp. z o.o. i wsp. Sp. Kom. Tapioca granulate was produced in Thailand by THAI WORLD IMPORT & EXPORT Co., Ltd. (Bangkok, Thailand) and distributed in Poland by KK Polska Sp. z o.o. Fermented tapioca was produced in Brazil and distributed by SUCOS DO BRASIL—PRODUCTOS LATINO GMB (Neuss, Germany) under the trademark YOKI™. When opened, the experimental material was kept in a tightly closed package, at room temperature, according to producers’ recommendations.

The chemical reagents used to maintain appropriate relative humidity conditions in the desiccators were high-quality, pure, analytical compounds. The water used to prepare the saturated solutions of these reagents was distilled water.

### 2.2. Methods

Differences in the physical properties of the analyzed starches were established by comparing value distributions of parameters characterizing the size and shape of their particles determined using a Morphology G3 automatic particle analyzer (Malvern Instruments, Malvern, UK). The analyzer enables determination of the size distribution of solid particles, with sizes ranging from 0.5 to 10,000 µm. Estimations were made for value distributions of such parameters as: diameter, circularity, convexity, elongation, shape coefficient, and solidity [22].

Water content was determined according to the Polish Standard (PN-ISO 712:2002) in the AquaLab apparatus series 4 model TE (Decagon Devices, Inc., Pullman, WA, USA), exact to ±0.003, at a temperature of 20 ± 1 °C.

Sorption isotherms were plotted using the static-desiccator method at ambient temperature of 20 ± 1 °C, in a water activity (*a_w_*) range from 0.07 to 0.98, and the time (30 days) required to reach the dynamic equilibrium between the analyzed samples and saturated solutions of respective substances was: NaOH·H_2_O (0.0698); LiCl·H_2_O (0.1114); KC_2_H_3_O_7_·1.5H_2_O (0.231); MgCl_2_·6H_2_O (0.3303); K_2_CO_3_·2H_2_O (0.440); Na_2_Cr_2_O_7_·2H_2_O (0.548); KJ (0.6986); NaCl (0.7542); KCl (0.8513); KNO_3_ (0.932); and K_2_Cr_2_O_7_ (0.9793). Equilibrium water contents were determined based on the initial masses of the analyzed samples with established water content, and then their changes triggered by the incubation process in desiccators. The values achieved allowed plotting of the isotherms of water vapor sorption in the tested range of water activities. Each point on each of the plotted isotherms was the arithmetic mean from three parallel determinations. Differences in the course of the sorption isotherms in the entire *a_w_* range were analyzed statistically using the Student’s *t*-test of differences between mean values for dependent variables. Differences were considered statistically significant at *p* < 0.05.

The course of sorption isotherms was analyzed mathematically using the *BET* Equation (1):(1)$$v=\frac{v_{m}\cdot c_{BET}\cdot a_{w}}{\left(1-a_{w}\right)\cdot \left[1+\left(c_{BET}-1\right)\cdot a_{w}\right]}$$
where:
*v*—adsorption, g H_2_O/100 g d.m.;*v_m_*—maximal adsorption value corresponding to the complete coverage of the surface with a monomolecular layer of the adsorbate, g H_2_O/100 g d.m.;*c_BET_*—energy constant *BET*, describing the difference between the chemical potential of crude adsorbate molecules and those in the first adsorption layer, kJ/mol;*a_w_*—water activity at the adsorption temperature [23].


The analysis also used the *GAB* Equation (2):(2)$$v=\frac{v_{m}\cdot c_{GAB}\cdot k\cdot a_{w}}{\left(1-k\cdot a_{w}\right)\cdot \left(1-k\cdot a_{w}+c_{GAB}\cdot k\cdot a_{w}\right)}$$
where:
*v*—adsorption, g H_2_O/100 g d.m.;*v_m_*—maximal adsorption value corresponding to the complete coverage of the surface with a monomolecular layer of the adsorbate, g H_2_O/100 g d.m.;*c_GAB_*—energy constant *GAB*, describing the difference between the chemical potential of adsorbate molecules in the first adsorption layer and higher layers, kJ/mol;*k*—constant correcting properties of multilayer molecules compared to the liquid phase;*a_w_*—water activity at the adsorption temperature [23].


The characterization of the sorption properties, in terms of hygroscopicity, using the *BET* and *GAB* models consisted of determining the maximal adsorption value corresponding to the complete coverage of the surface with a monomolecular layer of the adsorbate, called the monolayer (*v_m_*). Estimation of the monolayer allowed computing of the specific adsorption area (*a_sp_*) using Equation (3):(3)$$a_{sp}=\omega \cdot \frac{v_{m}}{M}\cdot N$$
where:
*a_sp_*—specific sorption area, m^2^/g;*N*—Avogadro number, 6.023 × 10^23^ molecules/mol;*M*—molecular weight of water, 18 g/mol;*ω*—water cross-section area, 1.05 × 10^−19^ m^2^/molecule [24].


In addition, energy constants *c_BET_* and *c_GAB_* were determined, and the *k* constant in the *GAB* equation was estimated.

Sizes and volumes of capillaries of the examined starch samples were determined for the area of capillary condensation using Kelvin’s Equation (4), assuming the cylindrical shape of the capillaries.
(4)$$lna_{w}=-\frac{2\cdot \sigma \cdot V}{r_{c}\cdot R\cdot T}$$
where:
*V*—molar volume of the liquid, g/mol;*σ*—surface tension of the liquid, N/m;*R*—universal gas constant, J/(mol K);*T*—temperature, K;*r_c_*—capillary radius, nm [24].


The *BET* equation parameters were identified based on empirical data in the water activity range of 0.07 < *a_w_* < 0.50 [25]. In turn, *GAB* equation parameters were identified based on empirical data from the entire *a_w_* range studied [26].

### 2.3. Statistical Analysis

The statistical significance of differences between the mean water content and water activity in the tested samples was measured using the ANOVA test and the Tukey post hoc test.

Parameters of *BET* and *GAB* equations were determined using a numerical algorithm based on non-linear regression and a Monte Carlo algorithm. Minimizing the residual sum of squares (*RSS*) was adopted as the target function. Standard errors of the equations’ determined parameters were estimated by a numerical algorithm using the SolverAid macro command, which reflects estimates of uncertainty of values of parameters obtained from Solver [24]. Finally, the fit of empirical data to both equations was characterized based on the evaluation of the residual sum of squares (*RSS*) Equation (5):(5)$$RSS=\sum {\left(v_{e}-v_{0}\right)}^{2}$$
and root mean square (*RMS*) error Equation (6) expressed in % [27,28]:(6)$$RMS=\sqrt{\frac{\sum {\left(\frac{v_{e}-v_{0}}{v_{e}}\right)}^{2}}{N}}\cdot 100\%$$
where:
*N*—number of data;*v_e_*—experimental equilibrium water content, g H_2_O/100 g d.m.;*v_o_*—predicted equilibrium water content, g H_2_O/100 g d.m. [28,29].


All computations were made in Excel 2019.

## 3. Results and Discussion

The first stage of the study involved a comparative analysis of the characteristics differentiating the parameters of cassava starch particles. This study stage aimed to identify the most effective applications of the starch products tested. In this case, the knowledge of physical properties is the second indicator, after the characterization of biological and chemical properties, enabling the rational management of the production process involving cassava starch, e.g., in food or pharmaceutical industries.

Hygroscopic properties represent a cumulative effect of many factors determining the affinity to water molecules [23]. These factors include, for instance, particle morphology and chemical composition [30]. The analyzed starches differed in their physical properties described by means of such parameters as: diameter, circularity, convexity, elongation, shape coefficient, and solidity (Table 1).

In terms of physical properties, the particles of the analyzed starches differed mainly in size, which was compared based on diameter. The fine-powdered starches (NS and FS) were polydisperse systems differing significantly in their particle diameters but with normal distributions in the values of this parameter. The diameter of fermented starch (FS) particles was greater than that of native starch (NS). This observation was also confirmed by higher values of solidity parameter recorded in the case of FS starch. In turn, the particles of starch granulate (SG) had significantly greater diameters than the two other starches and featured a bimodal distribution of the values of this parameter. The particles of the fine-powdered starches (NS and FS) showed greater variability in their shape regularity. Although the mean values of the circularity parameter pointed to their more remarkable similarity to a square (0.871 ÷ 0.895) compared to starch granulate (SG) particles (0.801), the comparison of the range of circularity values indicated the opposite. Among the particles of the fine-grained powders, there were also perfectly circular particles (1.000) as well as those not circular at all (0.028 ÷ 0.036). In contrast, there were no perfectly circular particles (0.965) of the starch granulate, but even the least circular particles of this starch type were more regular (0.417) than those of the fine-powdered starches (NS and FS). Analogous observations were made for the convexity values of the cassava starch particles. The particles of fine-powdered starches (NS and FS) were similar in terms of distribution of convexity value. In contrast, the starch granulate (SG) particles were characterized by greater clustering of their values manifested in a narrower range of the extreme values. By analogy, the values of the parameter described as elongation fitted within a substantially broader range in the case of SG particles compared to the NS and SS particles. As a consequence, the shape coefficient values of SG particles also fell in a significantly narrower range than those determined for the particles of the two other starch types. It may be concluded that the starch particles in the form of granules (SG) differed significantly from those of the fine-powdered starches (NS and FS), which also differed significantly in their sizes (diameter and solidity). The starch granulate (SG) differed significantly in terms of its particle characteristics from the two other starches (NS and FS). It may also be concluded that the fermentation process applied to produce fermented starch modified the size of its particles, including their diameter and solidity values. In contrast, it did not modify the shape, circularity, convexity, or elongation of cassava starch particles (Table 1).

A comparison of native cassava starch (NS) to other starches shows that, in terms of the diameter of its particles, it is very similar to native potato starch. However, its particles are more solid (massive) than those of potato starch [31].

The hygroscopic properties represent a cumulative effect of: interactions between body surface and water, water vapor condensation in capillaries, concentration and type of water-soluble substances, and water content of a product [32]. Water significantly affects the physical, chemical, and biochemical properties and microbiological safety of a biomatrix. In addition, it determines its susceptibility to degradation [31]. Hence, the water content and activity of the analyzed cassava starches were determined to compare their hygroscopicity (Table 2).

Data collated in Table 2 show that the starch granulate (SG) had the lowest content and, consequently, the lowest water activity, probably due to its preparation method (loose starch compression). On the other hand, more minor differences in water content and activity were found between the fermented starch (FS) and native starch powder (NS). In addition, the water-content-to-water-activity ratio enables conclusion of a greater affinity of water to fermented starch (FS).

The statistical assessment of the significance of differences between the mean values of water content and activity performed with the ANOVA test and the Tukey post hoc test (T) showed that only native and fermented starch did not differ significantly in terms of water content (T_crit._ = 0.868; T_NS/SG_ = 4.2417; T_NS/FS_ = 0.8223; T_SG/FS_ = 3.4193). On the other hand, the assessment of the differentiation in the water activity level showed statistically significant differences between all the samples (T_crit._ = 0.0081; T_NS/SG_ = 0.1125; T_NS/FS_ = 0.0357; T_SG/FS_ = 0.0768).

Despite significant water content differences, each starch sample’s water activity was low enough to ensure their microbiological stability. Microorganisms with the most minor demands in terms of water availability require water activity over 0.6 to initiate their vital functions [33].

Valuable information on the state of water in the material is provided by sorption isotherms, which enable establishment of product sensitivity to water in the form of vapor and the degree of water absorption by this product, as well as predicting changes in the material during storage affected by water availability [34]. In addition, the shape of an adsorption isotherm enables identification of a characteristic water-binding mechanism in a given material [35,36]. Figure 1 presents sorption isotherms of the analyzed cassava starch products.

Sorption isotherms of the analyzed starches had a sigmoidal shape and continuous course across the entire range of water activities tested. This indicates that water adsorption by starch caused no changes in the structure of starch granules, and, in particular, it did not lead to the crystallization of components [35].

The sorption isotherm of the starch granulate (SG) differed significantly from the other starches in terms of its position in the reference frame. Such a position of the isotherm indicates that, during storage of the analyzed starches in the environment with a specified humidity level, starch granulate (SG) will adsorb significantly less water reaching the same level of its activity as the other two fine-powdered starches (NS and FS). In addition, if all analyzed starches have the same water content, the starch granulate (SG) will feature the highest water activity. This, in turn, will make it the least stable, primarily regarding microbiological safety. The most likely reason behind the significantly different hygroscopic properties of SG is its smaller specific surface area compared to the developed surface of fine-powdered starches.

The second inflexion on the sorption isotherms of all starch samples was observed at *a_w_* ≈ 0.7. Suriyatem and Rachtanapun [37] reported similar findings. This characteristic point indicates the intensification of the sorption process due to the initiated phenomenon of capillary condensation [38]. This phenomenon consists of the filling of capillaries on the surface of a solid body with water molecules, leading to the modification of its properties to resemble those of free water [31].

The course of sorption isotherms in the entire *a_w_* range was also compared in the statistical analysis using the Student’s *t*-test for differences between mean values for dependent pairs [22,31]. The isotherms of all analyzed starches differed significantly in their course (t_SN/FS_ = 3.3096, t_SN/SG_ = 6.7958, t_FS/SG_ = 6.9045, t_crit._ = 2.228). The determined statistical values confirm a huge difference between the hygroscopic properties of starch granulate (SG) and those of the other two starch samples. In addition, they show that the fine-powdered starches (NS and FS) also differed in their hygroscopicity. The above results allow concluding that both starch fermentation and compression into compact granules (granulation) lead to significant differences in its hygroscopic properties. This is an important finding, as certain works demonstrate a paucity of data that would confirm the correlation between the physical properties (size and shape) and hygroscopic properties of powders or small-sized solid bodies [39]. The results of the research conducted by Ikhu-Omoregbe [40] showed that the fermentation of cassava starch causes a significant change in sorption properties. Furthermore, the results of many other studies have shown that starch fermentation changes the sorption properties of cassava starch. At the same time, it does not cause changes in the differentiation in the course of adsorption and desorption (hysteresis loop) [21].

The empirically determined sorption isotherms were described using two theoretical mathematical models: *BET* and *GAB* [36]. The *BET* model is most frequently employed to describe the structure of a product and sorption phenomena. It assumes that the shape of a sorption isotherm is due to the complex character of the sorption process on porous bodies [39]. Using the simplifying assumptions described in the literature, this model estimates the water-bound content in the so-called monolayer [37]. When making these estimates, caution should be exercised regarding the model’s limitation resulting from a tendency for significant overestimation of the predicted results in the range of high *a_w_* values [39]. Table 3 presents the parameters of the *BET* equation of the analyzed cassava starches.

The goodness of fit of empirical data to the results generated using the *BET* model was evaluated based on the residual sum of squares (*RSS*) and root mean square (*RMS*) errors. The comparison of the *RSS* values shows that the *BET* model described empirical data of all starches with similar accuracy. In turn, the *RMS* value determined for SG exceeded the threshold of 10%, indicating a good fit of the model to experimental data [41]. This means that the *BET* model better describes the hygroscopicity of fine-grained powders than granules; hence, the results characterizing the fine-grained powders can be claimed to be more reliable.

The monolayer (*v_m_*) describes the sorption capacity of the adsorbent and the availability of polar sites to the water vapor [36]. Karel [42] demonstrated that the monolayer of various natural products ranged from 4 to 11 kg H_2_O per 100 kg dry matter. The *v_m_* values estimated for cassava starches fitted within this range. The *v_m_* values estimated for individual starches differed, indicating that both cassava starch fermentation and granulation affected its water-binding capability. The fermentation process modified the physical parameters of native, which means a crystalline form of starch, having sparse amorphous regions due to peripheral damage of starch granules [35]. As a result, starch particles became more solid, and the monolayer capacity decreased. In turn, upon the granulation process, the physical properties of starch particles changed radically (Table 1), as did the monolayer capacity.

Knowledge of the monolayer is essential for designing processes of food drying and storage, and also for ensuring appropriate conditions during its transport [43]. Furthermore, the maximum permissible water content in the product is determined in practice based on the monolayer estimation [44]. Given the above, it may be concluded that the native cassava starch (NS) is less sensitive to water content changes in the environment compared to the fermented (FS) or granulated (SG) starches. When the monolayer capacity is low, the lesser amount of water adsorbed from the environment will result in the critical humidity. The latter, in turn, will trigger undesirable physical modifications, i.e., product caking and hardening, and especially hazardous microbiological changes [45].

The energy constant *c_BET_* informed about the energy released during sorption. Its values determined for all analyzed starch were low, pointing to the similar course of the sorption phenomenon and its physical nature. The enthalpy value approximating 20 kJ·mol^−1^ usually does not affect the nature of physically adsorbed molecules [46]. Notably, the highest load of energy was released during the adsorption of water molecules on the surface of starch granulate.

Ocieczek and Mesinger [31] conducted an analogous study with the numerical method of estimating *BET* model parameters for gluten-free wheat, maize, and potato starches. A comparison of the present study results with the findings from their study shows that cassava starch has a significantly larger monolayer. In turn, the sorption process in all these starches entails similar energetic transformations.

The second model tested in the study was the *GAB* model. Table 4 presents the parameters of the *GAB* equation of the analyzed cassava starches. Its use to study the sorption properties of dry products spurs a growing interest, especially among food technologists [36]. The *GAB* model was based on the *BET* theory by taking into account the modified properties of an adsorbent in terms of multi-layer adsorption [47,48]. It enables description of sorption isotherms in almost the entire *a_w_* range (from 0.05 to 0.93) and extrapolates data obtained at different temperatures [49]. Due to the above, this model may be applied in calculations used in product logistics management.

Lewicki demonstrated that maintaining the calculation error of the monolayer estimated based on the *GAB* equation at ±15.5% required the *k* constant to fit the range of 0.24 ÷ 1, and the *c_GAB_* constant to be higher than 5.67 [50]. These conditions were met in each of the analyzed cases. In addition, the estimated *RSS* and *RMS* values indicate that the *GAB* model proved very good in describing water vapor sorption by the analyzed starch samples. Model parameters were calculated for each starch type with the same accuracy. In turn, a comparison of *RMS* values shows that the *GAB* model proved better than the *BET* model in describing the sorptive properties of cassava starches [36].

The monolayer values estimated using the *GAB* equation were higher than those determined with the *BET* equation; however, the distribution of *GAB* model results was analogous to that achieved with the *BET* model. Native starch (NS) had the largest monolayer (11.2), whereas starch granulate (SG) had the smallest monolayer (8.7), which was most likely due to the modifications in the physical structure of starch granules [51] caused by fermentation or granulation.

The energy constant *c_GAB_* was defined as a difference between the enthalpy of adsorbate molecules in the first adsorption layer and the higher layers [24]. In turn, the strong exothermal interactions between the solid body matrix and water molecules were ascribed to a reduced process temperature and increased *c_GAB_* value [52]. Given the above considerations, it may be speculated that water sorption by the analyzed starch samples was physical in nature.

The *k* parameter also describes the deviation between desorption enthalpy and liquid adsorbent evaporation enthalpy and corrects the properties of molecules building the monolayer compared to the liquid phase. Such a deviation does not occur only when *k* equals 1 [49]. In addition, as Caurie [53] claims, the *k* value enables distinction between monolayer (*k* ≤ 0.5) and multilayer (*k* > 0.5) adsorption. For this reason, the determined values of *k* make the *GAB* model a reliable tool for describing the sorption properties of the analyzed starches. Furthermore, the *k* constant values determined for individual starch types were similar, which explicitly confirms the similarity of the examined material (cassava starch), which has earlier been pinpointed by Chirife and Iglesias [54]. Therefore, it may be concluded that the energy status of the molecules building multilayer systems on particular starch samples was very similar [26].

The last stage of the study aimed to determine selected parameters describing the microstructure of the surface of the analyzed starches. Table 5 presents the parameters describing the microstructure of the surface of the cassava starches. The first determined parameter was the specific sorption area, which is a derivative of the monolayer. Hence, the results obtained were analogous to those describing the monolayer. The second parameter was the total volume of capillaries, describing the total volume occupied by water as a result of filling the micro-, meso-, and macrocapillaries. Finally, the third parameter was the capillary radius that was filled as a result of initiating capillary condensation [55].

The results obtained indicate that the native (NS) starch, and therefore semicrystalline starch, was characterized by the most developed specific sorption surface area. As a result of the fermentation process and granulation, the specific surface of the starch decreased. This change, however, cannot be associated with an increase in the ordering of starch particles, but only with their physical modifications. Starch fermentation resulted in the increased solidity of its particles. However, as a result of granulation, all parameters describing the physical state of starch particles changed radically (the diameter, solidity, circularity, convexity, and shape coefficient increased radically, whereas elongation decreased) (Table 1). Moreover, the granulation of cassava starch contributed to the reduction in the diameter and total volume of capillaries filled during capillary condensation. Fermentation did not cause significant changes in the surface microstructure parameters.

Cassava starch-based products can complement the assortment of starch products available on the Polish market. Native cassava starch is similar to native potato starch in terms of particle characteristics but significantly different in this aspect from native wheat gluten-free starch and native maize starch [31].

Edible cassava starches, regardless of their type (native, fermented, granulated), showed strong hygroscopic properties [49], which were described and compared graphically using sorption isotherms and also mathematically based on the parameters of sorption models.

Sorption isotherms plotted for all tested cassava starches (NS, SG, FS) were sigmoidal and continuous over the entire range of water activities. This finding justified using *BET* and *GAB* models to identify the parameters of hygroscopicity characteristics.

The mathematical sorption models (*BET* and *GAB*) used to explore experimental data were based on well-established theoretical foundations [39]. These models described the experimental data very well, which was confirmed by the low *RSS* and *RMS* values, and by errors with which the parameters of both equations were estimated. However, taking into account the recommendations of the European COST 90 Project [56], a broader range of experimental data used to determine the parameters of the *GAB* model, the possibility of transferring the results obtained using the *GAB* model to different temperatures, and slightly better results of this model fitting to the original data, it can be concluded that the *GAB* model should be recommended for the description and comparison of the hygroscopic properties of all tested cassava starches.

## 4. Conclusions

The analysis of the results obtained using both models indicates that:Native cassava starch significantly differs from potato starch in terms of hygroscopicity described by the parameters of the *BET* model, despite similar particle size characteristics.Modification of cassava starch, both through fermentation and granulation, significantly modifies its physical and hygroscopic parameters.Modification of physical/chemical properties, and consequently hygroscopic properties of cassava starches, opens avenues for its more comprehensive and targeted use in both the food and pharmaceutical industries.Modification of cassava starch may contribute to the rational management of the food or pharmaceutical production processes, considering the diverse needs of these industries and consumers.

## Figures and Tables

**Figure 1 foods-11-01966-f001:**
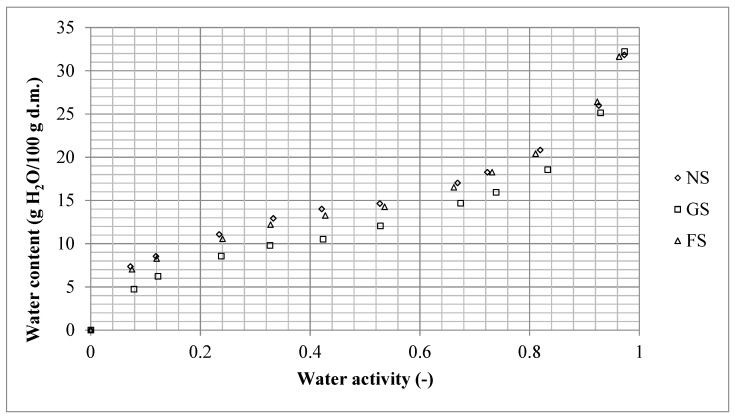
Comparison of water vapor sorption isotherms of native starch (NS), starch granulate (SG), and fermented starch (FS) at a temp. of 20 °C. Source: results of the present study.

**Table 1 foods-11-01966-t001:** Selected physical characteristics of the analyzed cassava starches.

Characteristics of Value Distribution	Parameter					
	Min.	Max.	Mean ± SD	D[*n*, 0.1]	D[*n*, 0.5]	D[*n*, 0.9]
Native starch (NS) (particles counted *n* = 214,733) (optic used: 10×)						
diameter, µm	1.1	145.3	12.7 ± 7.8	4.1	11.5	21.8
circularity	0.036	1.000	0.871 ± 0.147	0.666	0.922	0.990
convexity	0.338	1.000	0.983 ± 0.041	0.917	0.986	0.997
elongation	0.000	0.968	0.196 ± 0.153	0.025	0.162	0.416
shape coefficient	0.032	1.000	0.804 ± 0.153	0.582	0.834	0.971
solidity	0.089	1.000	0.974 ± 0.051	0.881	0.975	0.998
Starch granulate (SG) (particles counted *n* = 86) (optic used: 2.5×)						
diameter, µm	1297.6	3330.9	2465.6 ± 330.4	1283.0	1847.7	1862.4
circularity	0.417	0.965	0.801 ± 0.138	0.578	0.842	0.942
convexity	0.774	0.990	0.929 ± 0.057	0.843	0.946	0.982
elongation	0.001	0.515	0.131 ± 0.128	0.028	0.079	0.356
shape coefficient	0.485	0.999	0.869 ± 0.128	0.642	0.920	0.970
solidity	0.810	0.998	0.963 ± 0.038	0.904	0.976	0.992
Fermented starch (FS) (particles counted *n* = 317,432) (optic used: 10×)						
diameter, µm	1.1	380.8	13.3 ± 6.9	6.9	12.3	19.9
circularity	0.028	1.000	0.895 ± 0.145	0.697	0.959	0.993
convexity	0.329	1.000	0.984 ± 0.040	0.916	0.986	0.997
elongation	0.000	0.977	0.159 ± 0.145	0.019	0.104	0.376
shape coefficient	0.023	1.000	0.841 ± 0.145	0.622	0.893	0.978
solidity	0.177	1.000	0.976 ± 0.052	0.883	0.978	0.998

D[*n*, 0.1]—10% of the particles are smaller than this diameter. D[*n*, 0.5]—half of the particles are smaller than this diameter, and half are longer. D[*n*, 0.9]—90% of the particles are smaller than this diameter.

**Table 2 foods-11-01966-t002:** Water content and water activity of the analyzed cassava starch samples.

Product	Water Content [g H_2_O/100 g d.m.]	SD	Water Activity [−]	SD
Native starch (NS)	14.347	0.552	0.492	0.005
Starch granulate (SG)	10.108	0.081	0.380	0.001
Fermented starch (FS)	13.524	0.232	0.457	0.001

**Table 3 foods-11-01966-t003:** Parameters of *BET* equation determined for the analyzed cassava starches.

Parameter	Native Starch (NS)		Starch Granulate (SG)		Fermented Starch (FS)	
	Value	Error	Value	Error	Value	Error
*c_BET_*	0.8620	0.2610	1.1832	0.3646	0.9694	0.2741
*v_m_*	9.3989	3.7904	4.6637	2.0328	7.5257	2.9108
*RMS*	10.11		13.70		8.88	
*RSS*	6.6325	1.4869	4.7435	1.2574	5.7693	1.3868

**Table 4 foods-11-01966-t004:** Parameters of *GAB* equation determined for the analyzed cassava starches.

Parameter	Native Starch (NS)		Starch Granulate (SG)		Fermented Starch (FS)	
	Value	Error	Value	Error	Value	Error
*c_GAB_*	38.2050	4.6761	21.6854	1.8444	38.3936	4.0099
*k*	0.5732	0.0129	0.6535	0.0079	0.6182	0.0089
*v_m_*	11.1898	0.2784	8.7150	0.1690	10.3235	0.1913
*RMS*	7.71		9.64		7.23	
*RSS*	0.9842	0.3508	0.3867	0.2199	0.5681	0.2665

**Table 5 foods-11-01966-t005:** Microstructural characteristics of the surface of the analyzed cassava starches.

Product	Specific Sorption Area (m^2^/g d.m.)	Total Volume of Capillaries (mm^3^/100 g d.m.)	Capillary Radius Filled at *a_w_* = 0.7 (nm)
Native starch (NS)	393.1	122.4	2.18
Starch granulate (SG)	306.2	109.3	1.84
Fermented starch (FS)	362.7	121.6	2.13

## Data Availability

Data is contained within the article.

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
