# Peer review of "Hygroscopic Properties of Three Cassava (Manihot esculenta Crantz) Starch Products: Application of BET and GAB Models"

_foods, 2022, doi:10.3390/foods11131966_

Round 1
Reviewer 1 Report
The manuscript “Hygroscopic properties of three starch products from cassava (Manihot esculenta Crantz) described using the BET and GAB model (foods-1770466 )” has been nicely written. But there is a significant requirement for the Introduction and Discussion. Somehow the overall structure is problematic and needs significant revision.
BET and GAB should be given in full names in the first use.
Introduction:
In general, you mentioned about the general aspects related to cassava in this part but what about some physicochemical properties etc. water activity, amylose/amylopectin etc.? You have to consider these in this section too, for emphasizing the importance of the hygroscopic behaviour. Production process, safety, and stability are all connected with these properties.
Page 2, Lines 54-59: I suggest not to use bulleting to mention these food groups.
You have not mentioned about any previous research on the hygroscopic behaviour cassava starch/flour?
Page 2, Lines 88-93: “The study results provided new information related to the 88 management of the production process, safety, and stability of these products. The study goal was achieved through the experimental determination of sorption isotherms of three selected starch products, the analysis of their BET and GAB theoretical models, and the evaluation of their particle size distribution” You should present your hypothesis and novelty of your work instead of summarizing your findings in this section.
The introduction should be completely revised.
Materials
In the title “2. Results” you talk about your Materials and Methods and Results together (Page 3, Lines: 118-131). Please reorganize. These sections should be separate.
Please simply write the producers and where it is produced regarding to your samples.
Please change the abbreviation for “fermented starch powder (sour starch, SS)”. Maybe you can use FS or “sour starch” all through the text instead of fermented starch.
Results and Discussion
Table 1: For your results the digit numbers should be consistent.
For the foodnote: “Source: results of the present study” please remove this expression.
Table 2: 4 digits of sensitivity is too much.
You should add some more studies of fermented ground cassava and other starch samples. Discussion about the effect of fermentation on your findings is missing.
Conclusion
Please reformulate the conclusion. It should comprise your own ideas on your findings with future perspectives instead of other researchers’.
Author Response
We agreed to all Your comments. Thanks for Your time.
Reviewer 1
The manuscript "Hygroscopic properties of three starch products from cassava (Manihot esculenta Crantz) described using the BET and GAB model (foods-1770466 )" has been nicely written. But there is a significant requirement for the Introduction and Discussion. Somehow the overall structure is problematic and needs significant revision.
BET and GAB should be given in full names in the first use.
Answer: Thank You, it has been corrected, and abbreviations have been replaced with full names.
Introduction:
In general, you mentioned about the general aspects related to cassava in this part but what about some physicochemical properties etc. water activity, amylose/amylopectin etc.? You have to consider these in this section too, for emphasising the importance of the hygroscopic behaviour. Production process, safety, and stability are all connected with these properties.
Answer: Thank You for this very good tip. In the "Introduction", a paragraph (now 7) was added that discusses the role of carbohydrates, including the ratio of amylose to amylopectin as a factor determining the sorption of this starch. In addition, the crystalline state of starch granules in the raw material was related to the amount of water by recalling source data about water activity. It was also indicated how, according to the source materials, the processing of the raw material affects its sorption properties. Thus, the research problem analysed in this work was indicated.
Page 2, Lines 54-59: I suggest not to use bulleting to mention these food groups.
Answer: Thank You, the note was used in the text and bullets were removed, replacing them with uniform text, and separating phrases with semicolons for clarity.
You have not mentioned about any previous research on the hygroscopic behaviour cassava starch/flour?
Answer: Thank You for this remark. In the 7th paragraph added to the "Introduction", the latest data from 2022 on the variation in hygroscopic properties of cassava products were recalled. Earlier sources were also referred to in this paragraph.
Page 2, Lines 88-93: "The study results provided new information related to the 88 management of the production process, safety, and stability of these products. The study goal was achieved through the experimental determination of sorption isotherms of three selected starch products, the analysis of their BET and GAB theoretical models, and the evaluation of their particle size distribution" You should present your hypothesis and novelty of your work instead of summarising your findings in this section.
Answer: Thank You for this very valuable comment. The last paragraph of the "Introduction" has been redrafted in connection with its indication. As a result, the tested cassava products were clearly defined, differentiating their properties. Furthermore, it was explained why the theoretical BET and GAB models were used to analyse the results. Moreover, it was indicated which physical parameters of the cassava products were identified and why the relationship between the physical and sorption properties of the tested products was sought.
The introduction should be completely revised.
Answer: Thank You for the comment we deem reasonable. Therefore, this chapter of the work has been rebuilt, organised and supplemented with additional information, the essence of which was explained in references to the previous comments of Reviewer 1.
Materials
In the title "2. Results" you talk about your Materials and Methods and Results together (Page 3, Lines: 118-131). Please reorganise. These sections should be separate.
Please simply write the producers and where it is produced regarding to your samples.
Please change the abbreviation for "fermented starch powder (sour starch, SS)". Maybe you can use FS or "sour starch" all through the text instead of fermented starch.
Answer: Thank You for this suggestion, which we consider perfectly valid. In the text, the chapter "Results" was divided into a separate chapter "Material and methods", in which the tested samples were characterised, giving their producers and place of production. The method of determining the entire work of starch after the fermentation process was standardised by assigning it the abbreviation FS. An additional subsection was introduced, which describes the statistical methods used in the analysis of the results of empirical research.
Results and Discussion
Table 1: For your results the digit numbers should be consistent.
Answer: Table 1 presents the results with accuracy to three decimal places due to the low values of most of the presented physical parameters. Significant figures describing most of the parameters presented in this table appear only after the decimal point. The only exception was the results concerning the diameter of the tested particles, which were presented to one decimal place due to the high values of this parameter. Significant figures appear before the decimal point and are large numbers.
For the foodnote: "Source: results of the present study" please remove this expression.
Answer: The captions under the tables presenting the results of own research have been removed.
Table 2: 4 digits of sensitivity is too much.
Answer: The significant figures describing most of the parameters presented in this table appear only after the decimal point.
You should add some more studies of fermented ground cassava and other starch samples. Discussion about the effect of fermentation on your findings is missing.
Answer: In the text of the chapter "Results and discussion", a few sentences were added referring to the results of research by other authors in the field of fermented starch (FS) and its differentiation in terms of sorption properties compared to the dried native form with various particle sizes.
Conclusion
Please reformulate the conclusion. It should comprise your own ideas on your findings with future perspectives instead of other researchers'.
Answer: Thank You for this tip. Of course, we changed by referring in this chapter only to the findings from our research. All comments that have the character of discussions with the literature on the subject are presented in the chapter "Results and discussion".
Answer: Thank You for the valuable tips that have been included in the revised text.
Reviewer 2 Report
Interesting results and novelty work. A paper focuses on Hygroscopic properties of three starch products from cassava (Manihot esculenta Crantz) described using the BET and GAB model. Though the intention of the authors is highly commendable, there is lot of problems particularly in the presentation throughout the manuscript. Besides there are many grammatical mistakes throughout the manuscript, particularly in respect of use of singular and plural with the subject or verb. In view of the above comments, whole manuscript should be properly written and revised.
Abstract:
The abstract given here starts without any background for the present work. Of course, it contains brief details about experimental aspects and the obtained results. However this abstract does not follow the norm of an abstract, which should state briefly:
1. The purpose of the study undertaken, what are you trying to solve
2. brief mention of experimental aspects (without using abbreviations)
3. highlights of the results numerically
4. Important conclusions based on the obtained results
5. Potential applications
Therefore, it is suggested that the Abstract to be modified as per the suggestions given above.
Introduction
Introduction section is long with a many references based on the literature survey conducted by the authors. This is very good. However, this lacks in proper presentation of literature survey, which should have been systematic whereby existing scientific gaps should have been brought out. This should have given justification for the present study, which should be followed by the objectives of this study. In fact there is large amount of literature available on the characterization of starch. Similarly, a large number of methods to obtain these materials have been used mentioning their advantages and limitations. None of these have been brought out in this study whereby the authors have not justified why they have chosen the method they have used in their study. It should be noted that normally 'Introduction' should give brief background through literature survey for the study citing previous published work where-by scientific gaps that exist should be brought out. This would have led to justification for the present study. It is therefore suggested that ‘Introduction Section’ should be revised as suggested above because this Section is an important one from the point of view of taking up the present study.
Cassava is not cultivated in Poland due to unfavorable atmospheric conditions; hence the products made of it on an industrial scale have been introduced into the Polish market quite recently. So…. What are you want to explain? I think better to delete this sentence.
Relevant article on starch should be cited such:
Foods 2021;10:1609. https://doi.org/10.3390/foods10071609.
In my opinion the paper will have good merit if such applications can be demonstrated and reported. Can you give some example?
…… sorption isotherms and the parameters of selected mathematical models used to describe these isotherms. You want to study this thing but there is no information on this thing in your introduction section. Please do introduce.
Materials and Methods:
Normally, this section should have two main subsections. The first one is Materials which should give details of all materials used in the study, where from they were procured, known characteristics, if available (for e.g. cassava, where do you get it, what is the purity of the chemical and etc.).
The second subsection should be Methods, where methodologies used in the study should be given in a systematic way using sub section with numbers for each of the properties. First the processing or preparation aspects of the final material should be given followed by the characterization of prepared materials including preparation of samples for any specific property or morphology studies should be presented in a systematic way. Here one should also clearly mention the number of samples used, any standards followed for variety of properties, make and model of the instruments used for characterization, their accuracy and experimental conditions used, etc.
It should be known to the authors when one publishes any scientific paper, the results presented therein should be such they should be reproducible by any other person when the experiment is repeated using the same materials. In the present paper, it would be difficult for any other person to repeat the experiments because the chosen materials do not have any pre-history, which is required for other researchers to carryout experiments to check the possible reproducibility of the procedure adopted by these authors.
Some of the paragraph should be under results and discussion and if it is already there then this becomes repetition and hence can be deleted. Secondly, this Section is methods and hence only results should be mentioned and then it should be discussed preferably comparing it with earlier reported similar results by other researchers.
It is better to do some experiment on thermal analysis, dsc, density, water content and chemical compositions of the starch such as amylose, ash, far and protein. In my opinion the paper will have good merit if such properties can be demonstrated and reported as it shows fully potential of the starch.
Results & Discussion
Well written and easy for the reader to understand what the authors have conveyed.
Some of the paragraph should be under Methods and if it is already there then this becomes repetition and hence can be deleted. Secondly, this Section is Results & Discussion and hence only results should be mentioned and then it should be discussed preferably comparing it with earlier reported similar results by other researchers.
Throughout the manuscript, there are no comparison had been done with other published journal. Therefore, please support your statements with other researcher’s work in the section result and discussion. It should be discussed preferably comparing it with earlier reported similar results by other researchers.
How many sample did for each experiment? Please do ANNOVA test and standard deviation for all data collected and presented.
Although the mean values of the circularity parameter pointed to their greater
similarity to a square (0.871÷0.895) compared to starch granulate (SG) particles (0.801), the comparison of the range of circularity values indicated the opposite. Among the particles of the fine-grained powders there were also perfectly circular particles (1.000) as well as those not circular at all (0.028÷0.036). Please revise this section, perhaps the whole section. Why does you put divided symbol “÷”
A comparison of native cassava starch (NS) to other starches shows that, in terms of
the diameter of its particles, it is very similar to native potato starch, however, its particles are more solid (massive) than those of potato starch [28]. Please merge with the other paragraph.
Please revise Table 2… why do you separate the SD. Please do in one column.
Please combine the paragraphs that have only 2 sentences with other paragraph.
Figure 1. What is water activity unit?
Table 3 and 4 … what do you mean by error column. It is necessary to put in the Table.
Where is SD for Table 5.
Conclusions
Conclusions given here are do not reflect what had been achieved including many speculations. It is too long and should be in 1 paragraph. Hence these need to be suitably modified. It may be remembered that this Section forms a summary of all the major observations/ results obtained. Accordingly, here presentation should consist of the main Results or the observations of the study in short sentences probably with bullet points. This should stand alone or form a subsection of a Discussion or Results Section. Hence better to rewrite this Section based on the comments given in the whole text.
General Comments:
The paper though contains some interesting results and novelty work, it lacks in its proper presentation in the whole manuscript. Of course there is need for better language throughout the manuscript. It is suggested that the authors should take the help of native English speaking person to take care of this problem. It is suggested that the authors should revise the paper in the light of above comments/suggestions.
Author Response
Thank You very much for Your time. We had made a lot of changes as You suggested, eg. rewrited methodology section, apply ANOVA statistic, added some literature, rewrited conclusions. Unfortunately we can't develop abstract, because of the words limit.
Reviewer 2
Interesting results and novelty work. A paper focuses on Hygroscopic properties of three starch products from cassava (Manihot esculenta Crantz) described using the BET and GAB model. Though the intention of the authors is highly commendable, there is lot of problems particularly in the presentation throughout the manuscript. Besides there are many grammatical mistakes throughout the manuscript, particularly in respect of use of singular and plural with the subject or verb. In view of the above comments, whole manuscript should be properly written and revised.
Abstract:
The abstract given here starts without any background for the present work. Of course, it contains brief details about experimental aspects and the obtained results. However this abstract does not follow the norm of an abstract, which should state briefly:
- The purpose of the study undertaken, what are you trying to solve
- brief mention of experimental aspects (without using abbreviations)
- highlights of the results numerically
- Important conclusions based on the obtained results
- Potential applications
Therefore, it is suggested that the Abstract to be modified as per the suggestions given above.
Answer: Thank You for this proper remark and for indicating the way of redrafting the content of the abstract. We used the proposed modification method. Although the limitations on the length of the abstract did not allow for its more detail, thanks to the advice of Reviewer 2, we managed to make it more readable and methodically developed.
Introduction
Introduction section is long with a many references based on the literature survey conducted by the authors. This is very good. However, this lacks in proper presentation of literature survey, which should have been systematic whereby existing scientific gaps should have been brought out. This should have given justification for the present study, which should be followed by the objectives of this study. In fact there is large amount of literature available on the characterisation of starch. Similarly, a large number of methods to obtain these materials have been used mentioning their advantages and limitations. None of these have been brought out in this study whereby the authors have not justified why they have chosen the method they have used in their study. It should be noted that normally ‘Introduction’ should give brief background through literature survey for the study citing previous published work where-by scientific gaps that exist should be brought out. This would have led to justification for the present study. It is therefore suggested that ‘Introduction Section’ should be revised as suggested above because this Section is an important one from the point of view of taking up the present study.
Answer: Thank You for the indicated deficiency in the preparation of our work. The missing elements were supplemented with a discussion of the role of carbohydrates (the ratio of amylose to amylopectin) as a factor determining starch sorption. Sources were cited indicating that the crystalline state of starch granules and the presence of significant amounts of water determine the level of its activity as a factor of stability and safety. It was also cited how the raw material’s processing affects starch’s sorption properties. Thus, the research problem analysed in this work was indicated.
Cassava is not cultivated in Poland due to unfavorable atmospheric conditions; hence the products made of it on an industrial scale have been introduced into the Polish market quite recently. So…. What are you want to explain? I think better to delete this sentence.
Answer: We agree with the suggestion. This sentence has been deleted.
Relevant article on starch should be cited such:
Foods 2021;10:1609. https://doi.org/10.3390/foods10071609.
In my opinion the paper will have good merit if such applications can be demonstrated and reported. Can you give some example?
Answer: In the “Introduction” as well as in “Results and Discussions”, several new source items were cited, which allowed for the justification of selected research methods, and then for the interpretation of the results of own research and their discussion. At the moment, the conclusions are based on the results of own research presented in the context of knowledge from source materials.
…… sorption isotherms and the parameters of selected mathematical models used to describe these isotherms. You want to study this thing but there is no information on this thing in your introduction section. Please do introduce.
Answer: The current version explains why the theoretical BET and GAB models were used to analyse the results.
Materials and Methods:
Answer: Thank You for Your comment about this section, but we can only partially agree.Normally, this section should have two main subsections. The first one is Materials which should give details of all materials used in the study, where from they were procured, known characteristics, if available (for e.g. cassava, where do you get it, what is the purity of the chemical and etc.).
Answer: In the original manuscript, the research material was characterised by its origin (producer and place of acquisition). Therefore, it was not added.
The second subsection should be Methods, where methodologies used in the study should be given in a systematic way using sub section with numbers for each of the properties. First the processing or preparation aspects of the final material should be given followed by the characterisation of prepared materials including preparation of samples for any specific property or morphology studies should be presented in a systematic way. Here one should also clearly mention the number of samples used, any standards followed for variety of properties, make and model of the instruments used for characterisation, their accuracy and experimental conditions used, etc.
It should be known to the authors when one publishes any scientific paper, the results presented therein should be such they should be reproducible by any other person when the experiment is repeated using the same materials. In the present paper, it would be difficult for any other person to repeat the experiments because the chosen materials do not have any pre-history, which is required for other researchers to carryout experiments to check the possible reproducibility of the procedure adopted by these authors.
Answer: The second chapter, “Materials and methods”, has been divided into three sub-chapters. In this chapter, subsection 2.3 has been added, characterising the statistical methods that make it possible to compare the test results of individual material samples. It was also indicated in the text from the very beginning that the test was repeated in triplicate for each sample. Furthermore, the brands and models of instruments used in the research are given, as well as their accuracy and the conditions under which the experiments were conducted. Thus, we believe that our experience can be repeated without any problems. In addition, this chapter describes the class of reagents used to prepare desiccators for the determination of sorption isotherms.
Some of the paragraph should be under results and discussion and if it is already there then this becomes repetition and hence can be deleted. Secondly, this Section is methods and hence only results should be mentioned and then it should be discussed preferably comparing it with earlier reported similar results by other researchers.
Answer: We perceive this comment as unfounded. In the chapter on “Research material and methods”, in our opinion, there is no content that should be moved to the chapter “Results and discussion”. All the records presented in it were made synthetically and at the same time in detail so that any other researcher could recreate this study. In turn, the chapter “Results and discussion” referred to the interpretation and discussion of the results obtained in this work with the results of other researchers. This is the essence of the “Results and Discussion” chapter.
It is better to do some experiment on thermal analysis, dsc, density, water content and chemical compositions of the starch such as amylose, ash, far and protein. In my opinion the paper will have good merit if such properties can be demonstrated and reported as it shows fully potential of the starch.
Answer: Thank You for this valuable comment from Reviewer 2. We will include it in the next publication on cassava starch. In this work, we were interested in linking the results of research on selected physical parameters of the tested cassava starches with the hygroscopic properties of these starches. The relationship between the chemical composition and the chemical properties of starch appears attractive but should be described in a separate paper.
Results & Discussion
Well written and easy for the reader to understand what the authors have conveyed.
Some of the paragraph should be under Methods and if it is already there then this becomes repetition and hence can be deleted. Secondly, this Section is Results & Discussion and hence only results should be mentioned and then it should be discussed preferably comparing it with earlier reported similar results by other researchers.
Answer: In our opinion, there is no content in the “Results and Discussion” section that should be moved to the “Material and Methods” section. This chapter discusses in detail the results of the study of selected physical parameters of starch particles and their selected sorption parameters, which were developed based on two theoretical sorption models for reliability. In the context of these two datasets, relationships were analysed, and conclusions were drawn.
Throughout the manuscript, there are no comparison had been done with other published journal. Therefore, please support your statements with other researcher’s work in the section result and discussion. It should be discussed preferably comparing it with earlier reported similar results by other researchers.
Answer: Thank You for this suggestion, which we consider fully justified. Therefore, in the “Results and discussion” section, the results of the research and the findings of two studies were quoted, which showed that the fermentation of cassava starch causes a significant change in sorption properties. Considering that the second of the cited works is a meta-analysis, we concluded that we are referring to a wide range of systematised results and findings in this way. The results of these studies indicated, similarly to our findings, that starch fermentation changes the sorption properties of cassava starch. At the same time, these studies refer to the phenomenon of hysteresis loops, indicating that in this range, starch fermentation does not cause changes in the differentiation in the course of adsorption and desorption (hysteresis loops). We did not deal with this element of research on sorption, but we took it as a background for our own considerations.
How many sample did for each experiment? Please do ANNOVA test and standard deviation for all data collected and presented.
Answer: From the beginning, the text indicated that the test was repeated in triplicate for each sample. The test methods were described, indicating that each determination was performed three times, and the result was the arithmetic mean of these repetitions. The ANNOVA test and Tukey’s post-hoc test were performed, and the results are provided in the text.
Although the mean values of the circularity parameter pointed to their greater
similarity to a square (0.871÷0.895) compared to starch granulate (SG) particles (0.801), the comparison of the range of circularity values indicated the opposite. Among the particles of the fine-grained powders there were also perfectly circular particles (1.000) as well as those not circular at all (0.028÷0.036). Please revise this section, perhaps the whole section. Why does you put divided symbol “÷”
Answer: The paragraph after Table 1 synthesises the results presented therein, which were generated using the Morphology G3 automatic particle analyser (Malvern Instruments). In the methodical part of the work, reference was made to source materials, indicating where to find a detailed description to interpret such results. In our opinion, there is no place in this paper to describe the meaning of individual parameters and their ranges that indicate their interpretation. Our Work is not a work devoted to the operation and interpretation of results obtained with the Morphology G3 automatic particle analyser (Malvern Instruments). Using the symbol “÷” we indicate the range of values from-to “÷”.
A comparison of native cassava starch (NS) to other starches shows that, in terms of
the diameter of its particles, it is very similar to native potato starch, however, its particles are more solid (massive) than those of potato starch [28]. Please merge with the other paragraph.
Answer: Thank You for this comment. This paragraph has been merged with the previous paragraph.
Please revise Table 2… why do you separate the SD. Please do in one column.
Answer: The intended action of the authors of this paper was to present the results and the SD or standard error values in separate columns to emphasise their role in the interpretation of the data set.
Please combine the paragraphs that have only 2 sentences with other paragraph.
Answer: Unfortunately, we cannot agree with this opinion of Reviewer 2 because we believe that the text in the paragraph indicates a new thought. Therefore, it is not the length of the text but its substantive content that should determine the application of the paragraph. Using a paragraph, we wanted to emphasise certain essential elements of the text.
Figure 1. What is water activity unit?
Answer: Water activity is a dimensionless quantity, which results from the definition of this thermodynamic parameter, and therefore it is customary always to give its unit in the form [-].
Table 3 and 4 … what do you mean by error column. It is necessary to put in the Table.
Where is SD for Table 5.
Answer: Tables 3 and 4 present not only the estimates of the parameters of the BET and GAB equation but also the values of standard errors of the parameters of the equations, which were estimated through a numerical algorithm using the SolverAid macro command, which reflects estimates of uncertainty of values of parameters obtained from Solver. This description was presented in the methodical part of our work; therefore, we believe that it should not be repeated. In the case of the values presented in Table 5, it is impossible to estimate the SD value because these values are obtained by applying the analytical procedure described in the methodological part of our work. Please note the description of the procedure for applying equations 3 and 4.
Conclusions
Conclusions given here are do not reflect what had been achieved including many speculations. It is too long and should be in 1 paragraph. Hence these need to be suitably modified. It may be remembered that this Section forms a summary of all the major observations/ results obtained. Accordingly, here presentation should consist of the main Results or the observations of the study in short sentences probably with bullet points. This should stand alone or form a subsection of a Discussion or Results Section. Hence better to rewrite this Section based on the comments given in the whole text.
Answer: Thank You for the opinion with which we agree. Therefore, the chapter “Conclusions” has been redrafted, and as a result, it now includes only synthetic conclusions relating to the results of the research carried out by the authors.
General Comments:
The paper though contains some interesting results and novelty work, it lacks in its proper presentation in the whole manuscript. Of course there is need for better language throughout the manuscript. It is suggested that the authors should take the help of native English speaking person to take care of this problem. It is suggested that the authors should revise the paper in the light of above comments/suggestions.
Answer: Thank You for the constructive criticism from which we have concluded by making additions and corrections to the text of our article. We hope that its current form will be accepted.
Reviewer 3 Report
I reviewed the manuscript entitled, Hygroscopic properties of three starch products from cassava (Manihot esculenta Crantz) described using the BET and GAB 3 model. The concept of the research is good; however, the organization and presentation of research findings need to be improved.
The title can be re-written as Hygroscopic properties of three cassava (Manihot esculenta Crantz) starch products: Application of BET and GAB models
Use of BET and GAB models should be introduced in the section introduction. What is the need of using mathematical models on starch?
Include section 2.3. as statistical analysis
Line 94: is it results or methodology section? It seems methodology section
Line 86: write selected model names
There is no results section
Line 248: please remove
Table 2. perform statistical analysis to understand the differences
Abstract is poorly written. The findings of the study should be discussed in abstract.
Line 85: indicate the name of three cassava starch products
Table 5. remove line 400.
Line 88-89: please remove the line or move to at the end of introduction
Conclusions should be revised by highlighting the findings. Please avoid using references in conclusions. Conclusions are made based on research findings.
References are not according to journal format.
Author Response
Thank You very much for Your time and comments. We consider all of them, but there is one we cant complete. We can't add any information for the abstract, because there is words limit. Another comments we introduced.
Reviewer 3
I reviewed the manuscript entitled, Hygroscopic properties of three starch products from cassava (Manihot esculenta Crantz) described using the BET and GAB 3 model. The concept of the research is good; however, the organization and presentation of research findings need to be improved.
The title can be re-written as Hygroscopic properties of three cassava (Manihot esculenta Crantz) starch products: Application of BET and GAB models
Answer: Thank You for presenting the proposal, and we express the opinion that the proposed title will be better. Hence we have changed the title of our work.
Use of BET and GAB models should be introduced in the section introduction. What is the need of using mathematical models on starch?
Answer: Thank You for this remark, and we added a paragraph to the text devoted to the role of BET and GAB parameters in the study of the sorption properties of cassava starch.
Include section 2.3. as statistical analysis
Answer: We accept this comment gratefully. We supplemented the methodological part of the work by introducing chapter 2.3. In the discussion of the results, we included data from the statistical analysis of the results of primary empirical research.
Line 94: is it results or methodology section? It seems methodology section
Answer: Thank You for Your suggestion. We organized and expanded the methodical part of the work.
Line 86: write selected model names
Answer: The model names have been completed.
There is no results section
Answer: We consider Your suggestion justified. We have made some corrections in the text. The results section has been separated by separating it from the methodological section.
Line 248: please remove
Answer: The indicated line has been deleted.
Table 2. perform statistical analysis to understand the differences
Answer: We thank You for Your comments and consider them entirely justified. In the methodical part of the work, we added a subsection devoted to statistical analysis. This analysis has been performed, and its results have been substituted and discussed in the "Results and Discussion" chapter in the second paragraph after Table 2.
Abstract is poorly written. The findings of the study should be discussed in abstract.
Answer: We agree with the opinion of Reviewer 3. Therefore, the summary has been redrafted following the method indicated by Reviewer 2. However, it was not significantly expanded due to limitations concerning its length. However, we hope this version is a better description of the results of our work.
Line 85: indicate the name of three cassava starch products
Answer: The Reviewer's recommendation was taken into account.
Table 5. remove line 400.
Answer: The indicated line has been deleted.
Line 88-89: please remove the line or move to at the end of introduction
Answer: Thank You for Your attention. We have thoroughly edited this chapter and moved the indicated phrase to the end of the chapter.
Conclusions should be revised by highlighting the findings. Please avoid using references in conclusions. Conclusions are made based on research findings.
Answer: Thank You very much for this remark. The chapter "Conclusions" contains only the findings resulting from the conducted research. Other phrases have been moved to the "Results and Discussion" section.
References are not according to journal format.
Round 2
Reviewer 1 Report
Dear Editor,
I found the revisions as sufficient for acceptance.
Reviewer 3 Report
Based on the author’s response to reviewer comments/suggestions, the quality of the manuscript has improved. In my opinion, this version can be accepted for publication.
Suggestion to the authors-
Dear authors, please provide a point to point (with line numbers of the revised version of the manuscript) response to the reviewer’s comments in future, it will not only help to the reviewers, but also support the fast publication processing of your manuscript.